# Experimental Evidence of Specimen-Size Effects on EN-AW6082 Aluminum Alloy in VHCF Regime

**Stefano Invernizzi** [1,*,†] , **Francesco Montagnoli** [1,†] **and Alberto Carpinteri** [1,2,†]

1    Department of Structural, Geotechnical and Building Engineering, Politecnico di Torino, Corso Duca degli Abruzzi 24, 10129 Torino, Italy; francesco.montagnoli@polito.it (F.M.); alberto.carpinteri@polito.it (A.C.)
2    Department of Civil and Environmental Engineering, Shantou University, Shantou 515063, China
*    Correspondence: stefano.invernizzi@polito.it; Tel.: +39-011-090-4860
†    These authors contributed equally to this work.

**Featured Application: The work provides a robust methodology for the determination of fatigue life expectation in the VHCF regime of the EN-AW6082 aluminium alloy, taking into account the influence of specimen size. In this way, correct fatigue life expectation for EN-AW6082 aluminium alloy full-scale components can be obtained.**

**Abstract:** The present paper investigates the influence of the specimen size of EN-AW6082 wrought aluminium alloy subjected to very high cycle fatigue (VHCF) tests. The hourglass specimens were tested under fully reversed loading condition, up to $10^9$ cycles, by means of the ultrasonic fatigue testing machine developed by Italsigma® (Italy). Three specimens groups were considered, with a diameter in the middle cross-section ranging from 3 mm up to 12 mm. The stress field in the specimens was determined numerically and by strain gauge measurements in correspondence of the cross-section surface. The dispersion of experimental results has been accounted for, and data are reported in P-S-N diagrams. The decrease in fatigue resistance with increasing specimen size is evident. Theoretical explanation for the observed specimen-size effect is provided, based on Fractal Geometry concepts, allowing to obtain scale independent P-S*-N curves. The fatigue life expectation in the VHCF regime of the EN-AW6082 aluminium alloy full-scale components is rather overestimated if it is assessed only from standard small specimens of 3 mm in diameter. Experimental tests carried out on larger specimens, and a proper extrapolation, are required to assure safe structural design.

**Keywords:** ultrasonic fatigue test; aluminium alloy; size effects; VHCF; fractal geometry

## 1. Introduction

Since the introduction of the ultrasonic fatigue testing machine by Mason in 1950 [1,2], the interest of the scientific community in the long-lifetime of metallic materials in the very-high cycle fatigue (VHCF) regime has continuously increased [3]. In fact, due to the growing need to extend the lifetime of structural components, such as railway and wind-turbines components, offshore structure, and engine elements, the knowledge about the fatigue behavior in the VHCF domain has become of paramount importance in structural engineering applications [4]. More recently, the concept of very-high-cycle low-amplitude fatigue has been applied to the field of civil engineering to explain the collapse of the Morandi bridge (Italy) [5,6]. In fact, according to Invernizzi et al. [7] the failure of the stay cables of Polcevera viaduct may have been triggered by the combined effect of gigacycle fatigue and corrosion. Therefore, in order to reduce the testing time, suitable fatigue testing devices working at high frequencies have been designed, so that it is possible nowadays to perform tests up to $10^{10}$ cycles in a very short time in comparison with traditional testing techniques, such as servo-hydraulic or rotating bending machines [8]. In addition, another important issue regarding the VHCF field is the specimen-size effect on the fatigue

resistance in the ultralong life regime. In fact, VHCF tests are commonly performed on specimens with a standard diameter of 3 mm, so that the fatigue resistance of full-scale components has to be extrapolated through theoretical models, which nevertheless have not yet been fully validated experimentally by using ultrasonic fatigue testing machines. To this aim, Furuya [9] performed VHCF tests on high-strength steel specimens with a diameter in the middle cross-section ranging from 3 mm up to 8 mm. From these experimental campaigns, he found that the larger the risk volume, the lower the fatigue limit, where the risk volume is usually defined as the region where the stress is higher than the 90% of the peak value [10,11]. Moreover, Murakami et al. [12,13] found that the internal fatigue cracks origin is the largest defect within the risk-volume. In addition, the probability to find defects with larger sizes increases with the risk-volume, so that a decrease in the fatigue resistance in the VHCF region can be expected. A few years ago, Tridello et al. [14–16] designed an innovative specimen shape with a Gaussian profile, which can be used to enlarge the risk volume of the specimen. Therefore, they were able to test a high-strength steel, AISI H13 steel, and to analyze results from a wide range of different risk volumes from 194 mm$^3$ up to 2300 mm$^3$ [17,18]. More recently, in order to increase the range of tested risk volume, the same authors carried out VHCF experiments on AISI H13 steel hourglass and gaussian samples with risk volumes of 55 mm$^3$ and 5000 mm$^3$, respectively [19]. Tridello et al. [20,21] also investigated the VHCF response of SLM AlSi10Mg and SLM Ti6Al4V gaussian specimens with a large loaded volume, pointing out that specimen-size effects on the VHCF region are observed in non-ferrous metallic materials, like aluminum alloy, as well. Xue et al. [22] performed fatigue tests on Al-Si-Cu cast alloy specimens of different sizes beyond $10^9$ cycles with an ultrasonic fatigue testing machine operating at 20 kHz and $R = -1$. Firstly, they observed that pores within the material act as a preferential site for the fatigue crack initiation. Secondly, they observed experimentally that, for a certain number of cycles, the larger the risk volume of the specimen, the lower the fatigue life. In the present investigation, fully reversed ultrasonic fatigue tests up to $10^9$ cycles were performed on hourglass specimens made of EN-AW6082-T6 aluminum alloy, which is characterized by good tensile strength, weldability, and an excellent corrosion resistance. Furthermore, in order to investigate the specimen-size effects on this material, three different dimensions of specimens were considered, with diameter ranging between 3 mm and 12 mm. Firstly, the optimal geometry of the specimens was designed based on numerical finite element simulations. Secondly, strain gauge calibrations were carried out before the tests to validate the assumed stress distribution in the designed hourglass specimens. Furthermore, the experimental results were analyzed, and P-S-N curves were obtained for the three different specimens. Finally, by exploiting lacunar fractality concepts [23], the specimen-size effect on fatigue resistance was theoretically assessed, and scale invariant generalized P-S*-N curves provided.

## 2. Materials and Methods

### 2.1. Materials

The material object of the present experimental campaign is an EN-AW6082 wrought aluminum alloy received in T6 tempered condition. It is worth noting that, due to a good tensile strength and an excellent corrosion resistance, EN-AW6082 is one of the most popular alloys of the 6xxx Aluminium–Magnesium–Silicon family. The specimens used for the experimental tests were obtained through a machining process by two round bars of 20 and 30 mm in diameter, and 4 m and 3 m in length, respectively. In Table 1 the chemical composition of the tested aluminum alloy is reported.

**Table 1.** Chemical composition of the investigated EN-AW6082 aluminium alloy.

| Element | Si | Mg | Mn | Cu | Fe | Cr | Zi | Ti |
|---|---|---|---|---|---|---|---|---|
| Min% | 0.70 | 0.60 | 0.40 | | | | | |
| Max% | 1.30 | 1.20 | 1.00 | 0.10 | 0.50 | 0.25 | 0.20 | 0.10 |

Moreover, the physical and mechanical properties of the round bars made of aluminum alloy are reported in Table 2. The longitudinal dynamic elastic modulus, $E_d$, was measured with the impulse excitation technique on round bars of 6 mm in diameter with slenderness equal to 20, according to ASTM Standard E1876-15 [24]. The mass density, $\rho$, was determined by using an analytical weight balance characterized by an accuracy of $\pm 0.0001$ g. The ultimate tensile strength, $\sigma_u$, as well as the yield stress, $\sigma_y$, and the ultimate deformation, $\epsilon_u$, were determined on two dog-bone specimens for each round bar, which had been shaped according to ASTM Standard B557M-15 [25], by using a servo-hydraulic tensile test machine with 100 kN load cell. Finally, the Brinell hardness HB was measured for the two-round bars, according to ASTM Standard E10-18 [26]. The mean values and the corresponding standard deviations of the physical and mechanical properties are summarised in Table 2.

**Table 2.** Physical and mechanical properties of the investigated EN-AW6082 aluminum alloy.

| Diameter of Bar | $\rho$ (kg/m$^3$) | $E_d$ (GPa) | $\sigma_y$ (MPa) | $\sigma_u$ (MPa) | $\epsilon_u$ (%) | Brinnel Hardness (HB) |
|---|---|---|---|---|---|---|
| 20 mm | 2713 | 72.3 | – | – | – | $91 \pm 1.4$ |
| 30 mm | 2700 | 70.1 | $357 \pm 1.3$ | $375 \pm 0.2$ | $11 \pm 1.0$ | $91 \pm 1.7$ |

*2.2. Ultrasonic Fatigue Testing Machine*

The ultrasonic fatigue testing machine used in this work was developed by Italsigma® (UFTM MU90) and it is shown in Figure 1. The UFTM MU90 is equipped with an ultrasonic generator (Branson® DCX Series S 4 kW), which provides an electric signal at the piezoelectric transducer at 20 kHz (Branson® CR-20). Therefore, thanks to the converse piezoelectric effect, the latter converts the electric signal in a sinusoidal mechanical vibration with the same frequency of 20 kHz. In addition, it is possible to vary the amplitude of the mechanical vibration of the piezoelectric converter in the 5.3–21.7 μm range, by changing the output voltage setting of the ultrasonic generator. The piezoelectric converter is rigidly connected to a booster (Branson® 2000X Series Gold) through a screw connection, which provides fixed support to the whole mechanical system and amplification of the mechanical vibration equal to 1.5. Finally, a catenoidal horn (Branson® 126-192), characterized by an amplification factor of 2.0, is assembled in line with the booster with the aim of magnifying the displacement provided to the specimen, which is connected with an M6 screw. Therefore, the mechanical vibration is transmitted from the piezoelectric transducer to the specimen using the two mechanical amplifiers, so that an amplitude variable from 16 μm up to 50 μm is guaranteed at the free end of the specimen, where the maximum allowed value is evaluated in order to prevent unwanted failures of the horn. Furthermore, the mechanical vibration amplitude at the bottom of the specimen is monitored through an eddy current sensor (Micro-Epsilon® eddyNCDT 3300/3301) to keep the strain amplitude in the middle section of the specimen constant by adjusting the power setting of the ultrasonic generator. Notice that, since the piezoelectric transducer works in the 19.5–20.5 kHz range, all the components of the resonant system, as well as the specimen, must be designed in order to have an axial fundamental frequency included within the same interval. It is also worth noting that, when the fatigue crack propagates inside of the specimen up to a critical amount, the fundamental frequency falls below the lower limit of 19.5 kHz. Consequently, the test is automatically interrupted, since the piezoelectric transducer is no longer able to excite the mechanical components. Furthermore, an air-cooling system is used to control the temperature increment in the sample which could occur due to internal heat arising from high-speed deformations in ultrasonic fatigue tests [27]. At the same time, the temperature is monitored during the test with a pyrometer (Optris® CT LT22CF), which is characterized by an accuracy of $\pm 1$ °C, a resolution of 0.1 °C, and a measurable temperature range comprised between $-50$ °C and 975 °C. Besides, the ultrasonic fatigue testing machine (MU90) can operate discontinuously, in a pulse-pause mode, in order to

avoid excessive heating of the specimen. Finally, the UFTM is equipped with a load cell able to provide a maximum constant preload of 1.5 kN, so that it is possible to carry out tests with non-zero mean stresses ($R > -1$).

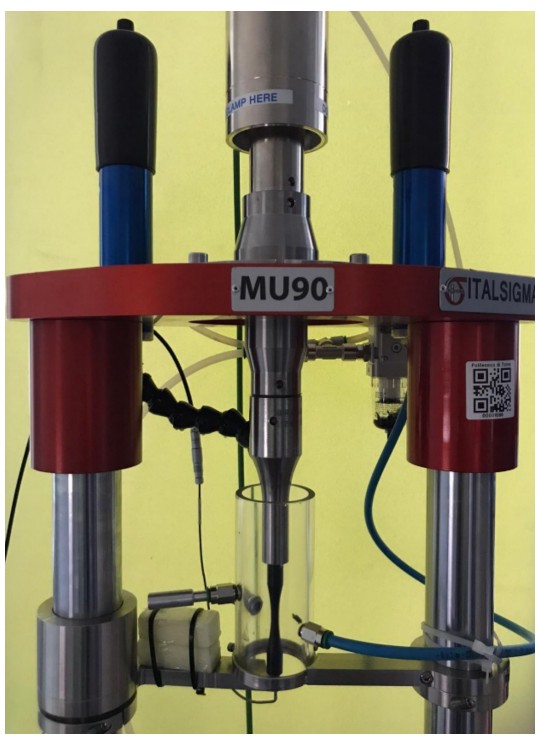

**Figure 1.** Ultrasonic fatigue testing machine MU90.

### 2.3. Specimens Design

In order to study the specimen-size effect on the fatigue resistance of aluminum alloy EN-AW6082, three different sets of hourglass specimens, respectively, with diameters of 3 mm, 6 mm, and 12 mm in the middle cross-section, were designed to be tested with the ultrasonic fatigue testing machine MU90. In particular, the specimens of 3 and 6 mm in diameter were obtained by the round bar of 20 mm in diameter, whereas the 30 mm round bar was used to obtain the largest specimens of 12 mm in diameter. The shape of the hourglass-shaped specimens was obtained, analytically, according to [3], as a function of the physical and mechanical properties of the adopted round bar. In addition, the normal stress $k_\sigma$ in the middle cross-section of the specimen subjected to unit amplitude mechanical vibration can also be assessed analytically. Subsequently, the obtained geometry for all three different specimens was checked by the commercial finite element code, ANSYS® Workbench. The axis-symmetry of the problem can be exploited to set up a 2D reduced model, discretized with 8-node axisymmetric quadrilateral elements, so that the computational cost of the analysis has been drastically reduced compared to a fully 3D finite element simulation. At first, a modal analysis was carried out to check that the resonance frequency of the specimens was as close as possible to the working frequency of the machine, i.e., 20 kHz. Subsequently, an harmonic analysis was performed to determine the stress concentration factor, $k_t$, which is evaluated according to [14]. In addition, the risk-volume has been evaluated, $V_{90}$, which is usually defined as the region where the stress amplitude is higher than the 90% of the peak value [9,11,28]. The above-mentioned dynamic mechanical properties of three different specimens, as well as the risk-volume, are summarised in Table 3, whereas Figure 2 shows a photograph of the specimens.

**Table 3.** Dynamic mechanical properties and risk-volume of the investigated three different specimens.

| Specimen | $k_\sigma$ (MPa/μm) | $f_{FEM}$ (Hz) | $k_t$ (–) | $V_{90}$ (mm$^3$) |
|---|---|---|---|---|
| 3 mm | 7.474 | 20,213 | 1.021 | 42 |
| 6 mm | 4.233 | 20,047 | 1.025 | 301 |
| 12 mm | 3.350 | 19,947 | 1.046 | 1731 |

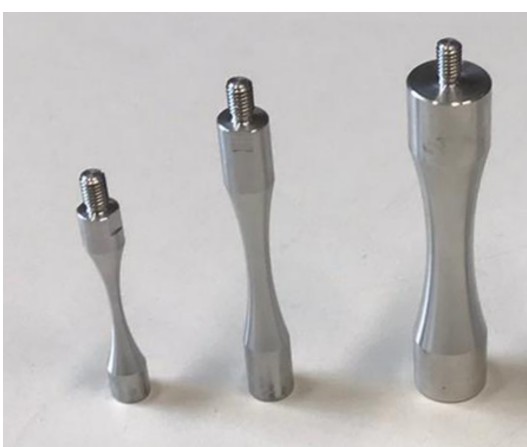

**Figure 2.** The three different investigated hourglass-specimens.

Table 3 shows that, the larger the diameter of the specimen, the lower the value of $k_\sigma$, whereas the stress concentration factor $k_t$ increases for larger specimen dimensions, while remaining less than 1.15. Moreover, it is relevant to observe that the largest specimen investigated in the present experimental campaign has dimensions larger than the ones commonly used for ultrasonic fatigue testing, which have the diameter in the middle-section ranging from 3 to 6 mm. On the other hand, the risk-volume obtained for the specimen of 12 mm in diameter is not so large if compared to the gaussian specimens tested in [17,19,29], although the diameter in the middle-section is similar enough. Tridello et al. [17,19,29], proposed to relate the specimen-size effect on the fatigue strength to the increase in the probability to find defects with larger size within the risk-volume, when the latter increases. Conversely, according to the present authors [23], the specimen-size effect is due to the lacunarity of the reacting cross-section, and can be explained by exploiting the concept of fractal medium. As a consequence, the decrease in the fatigue strength with the specimen size is not directly connected to the risk-volume, but rather to the specimen-characteristic size, e.g., the diameter, and to the fractal dimension of the middle cross-section. As mentioned above, the values of $k_\sigma$ for the specimens of 6 mm and 12 mm in diameter have been also validated experimentally by means of strain gauges measurements, which have confirmed the theoretically determined values reported in Table 3 and the numerical simulations. The specimens geometries used for the ultrasonic fatigue are shown in Figures 3a–5a, while in Figures 3b–5b the results of the numerical analysis with ANSYS Workbench are reported for the three different specimens. Eventually, in Figure 6a–c the photographs of the three specimens connected to the UFTM are reported.

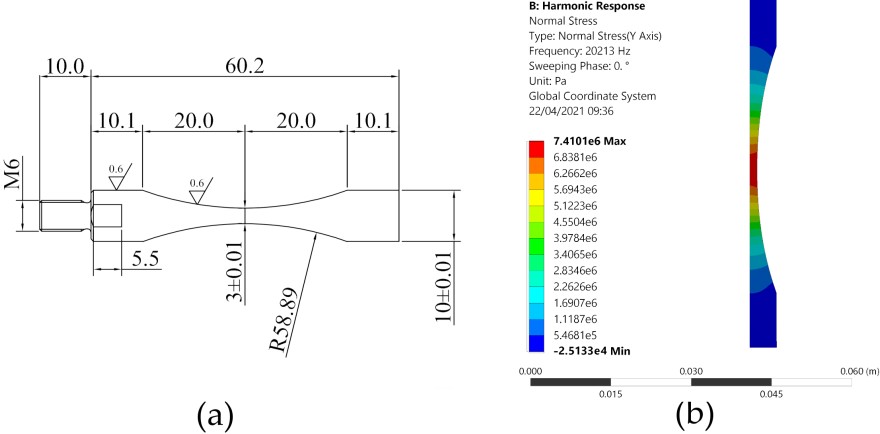

**Figure 3.** Specimen of 3 mm in diameter: (**a**) specimen geometry; (**b**) results of the numerical analysis.

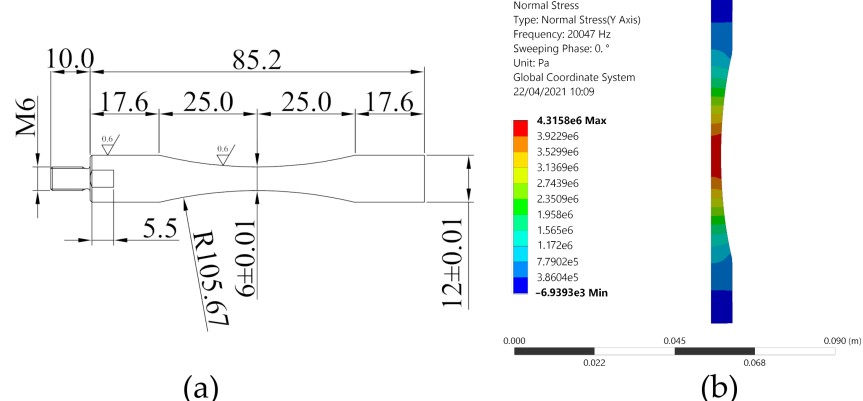

**Figure 4.** Specimen of 6 mm in diameter: (**a**) specimen geometry; (**b**) results of the numerical analysis.

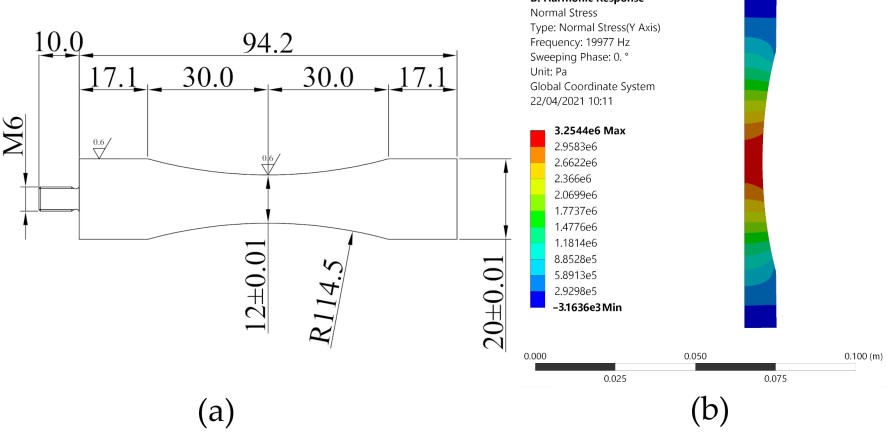

**Figure 5.** Specimen of 12 mm in diameter: (**a**) specimen geometry; (**b**) results of the numerical analysis.

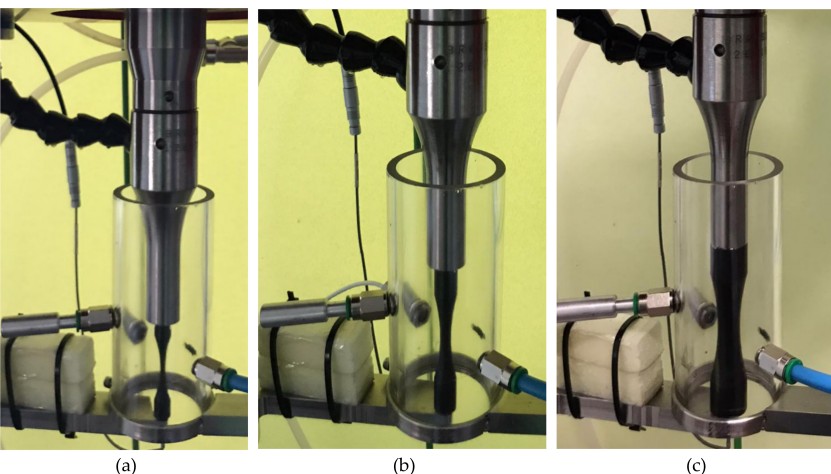

(a)          (b)          (c)

**Figure 6.** The three tested hourglass specimens: (**a**) 3 mm; (**b**) 6 mm; (**c**) 12 mm.

## 3. Results

### 3.1. Experimental Results and Fracture Surface Analysis

Ultrasonic fatigue tests were conducted under fully reversed constant stress amplitude tension-compression conditions ($R = -1$) at room temperature. The specimens were tested up to failure or up to a run-out of cycles equal to $10^9$. For the hourglass specimens of 3 mm and 6 mm in diameter it was possible to carry out the tests continuously, since the air-cooling system was able to keep the specimen temperature always below 50 °C, except for the very final stage of the total fatigue life, when the specimen temperature rose abruptly just before failure due to a crack propagation [27,30]. More in detail, the abrupt increase in the temperature due to the fatigue crack propagation lasts always less than 6–7% of the total lifetime of the specimens. For the specimen of 12 mm in diameter, to avoid heating of the samples, a pulse–pause sequence with a period of 1 s was applied for all loading levels. The pulse and pause lengths were set up at 500 ms and 500 ms, respectively. It is reasonable to suppose that the higher self-heating of the specimen of 12 mm in diameter is due to the larger sample dimensions, despite the investigated aluminum alloy is characterized by a higher thermal conductivity, compared to high-strength steel. In this way, the temperature of the specimens of 12 mm in diameter was kept below 50 °C for almost the duration of tests, except for the final stage of crack propagation in which the temperature reached in many cases a value higher than 100 °C. For all the three different sets of specimens, failures were obtained in the range between $7 \cdot 10^6 \div 10^9$ cycles, i.e., in the VHCF regime. In particular, sixteen tests were carried out on an hourglass-specimen of 3 mm in diameter at a stress amplitude ranging between a minimum value of 160 MPa and a maximum value of 210 MPa; thirteen of these led to a failure, whereas three were runouts. Moreover, seventeen tests were carried out on specimens of 6 mm in diameter, reporting two runouts at 150 MPa. Finally, fifteen failures were obtained for the specimens of 12 mm in diameter between 140 MPa and 180 MPa, whereas two runouts were obtained at 130 MPa. The experimental results are reported in Figure 7.

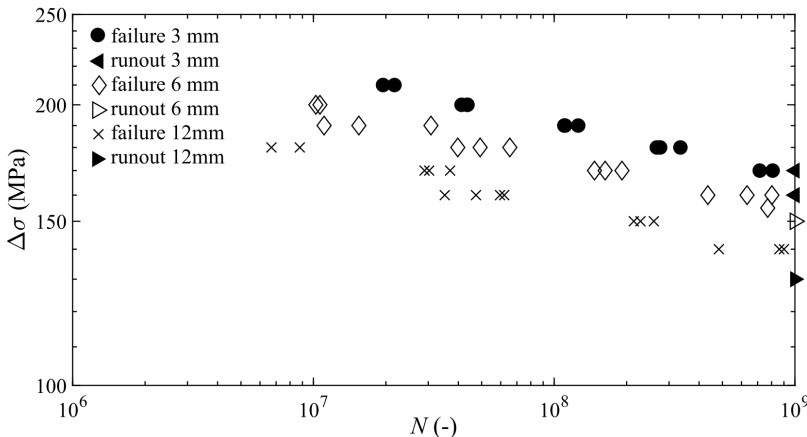

**Figure 7.** S-N diagram of the experimental data.

Figure 8 shows the fracture surface morphology of a specimen of 12 mm in diameter, which has been obtained by using Field Emission SEM. From the figure, it is possible to distinguish three different zones, i.e. the crack initiation origin, the crack propagation zone, and the final failure region. It can be observed that the crack initiation is located close to the sample surface, which is in accordance with other experimental investigations regarding aluminum alloys. It is worth noting that the total rupture of the specimen in two distinct parts was obtained with a three-point bending test, since the crack propagation during the ultrasonic test was such that the specimen frequency fell below the lower limit of 19.5 kHz, without yet leading to unstable crack propagation.

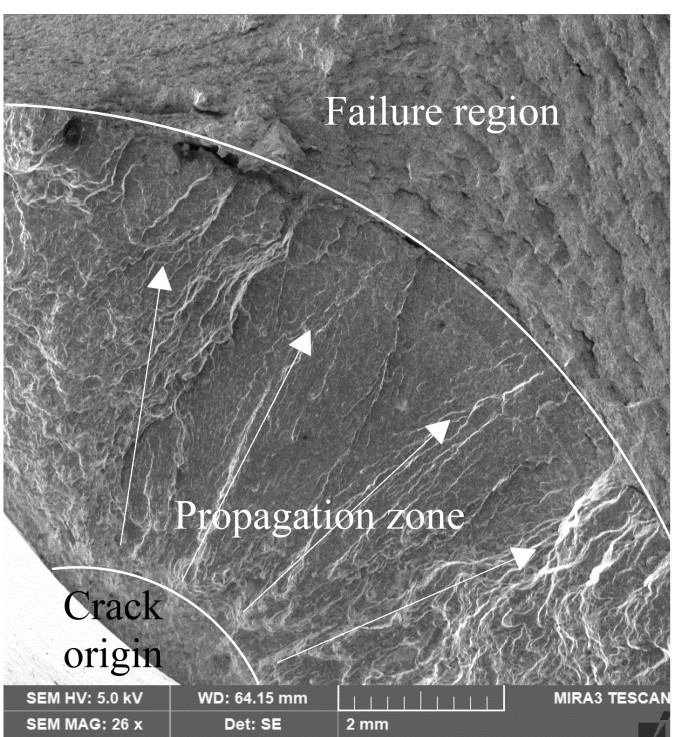

**Figure 8.** SEM image of the fracture surface morphology of the specimen of 12 mm in diameter.

Figure 9 shows a magnification of the crack propagation zone, where prominent striation marks with a river-like appearance are visible. Eventually, similar fracture morphology is observed for the other specimens of 3 mm and 6 mm in diameter.

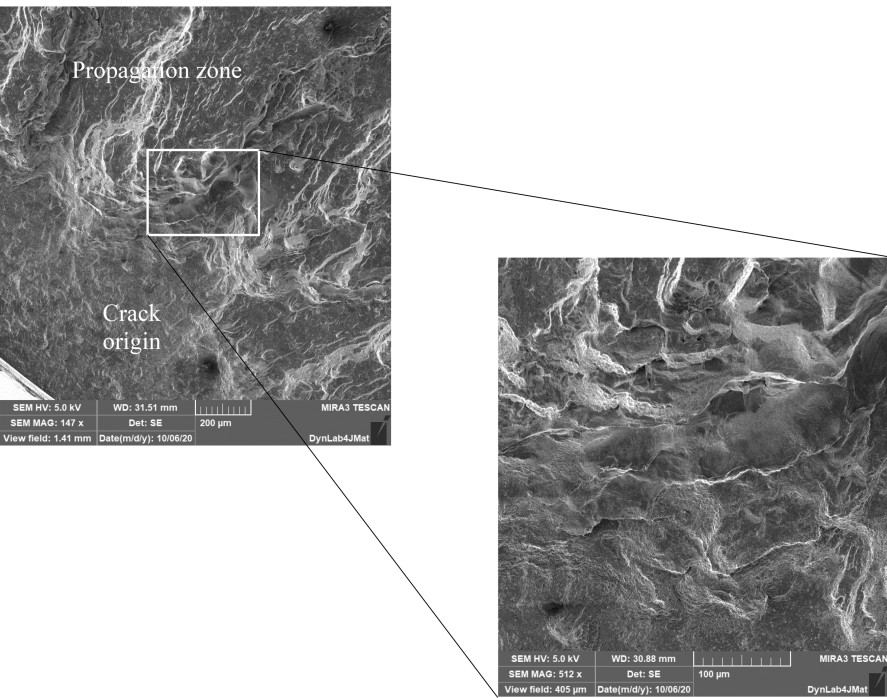

**Figure 9.** SEM images details of the propagation zone and striation marks of the specimen of 12 mm in diameter.

### 3.2. P-S-N Curves

In this sub-section, an estimation of the probabilistic S-N curves for the three different sizes is performed. In fact, fatigue test results are affected by a large scatter in the fatigue life, especially in the ultra-long fatigue regime [22]. Therefore, it is not sufficient to fit the experimental data with the least square method [31], since in this way only a mean life curve is obtained, whereas the influence of fatigue life statistical scatter is disregarded. In other words, it is strictly needed to use deterministic-stochastic models in order to provide reliable estimates of design S-N curves [32]. Since the aluminum cast alloys have no evident fatigue limit, the expression of the generic P-S-N curve can be properly interpolated by the following power-law equation:

$$N = \left( \frac{\Delta\sigma_0(P)}{\Delta\sigma} \right)^n \tag{1}$$

Equation (1), in the bi-logarithmic diagram, is represented by a straight line with the slope equal to $1/n$, and the intercept with the ordinate axis equal to $\Delta\sigma_0(P)$, which depends on the considered probability of failure. In other words, the fatigue life is considered as a stochastic variable described by the following two-parameter Weibull cumulative distribution function:

$$P(N, \Delta\sigma) = 1 - e^{\left[ -\left( \frac{N}{\beta} \right)^\alpha \right]} \tag{2}$$

The two constants $\alpha$ and $\beta$ are the shape and scale parameters, respectively, which are considered constant for a given stress range and a specimen size. It follows that, in order to obtain the probabilistic S-N curves, it should be necessary to fit the two parameters of the model with the available experimental data for each stress level. This procedure is not straightforward, and at least five samples should be considered at each stress range to provide a good estimate of the model parameters [32]. In the present experimental campaign, no more than three specimens for each stress range level are available. Fortunately, a specific statistical procedure proposed in [33,34] can be adopted in order to overcome the problem of limited available data. The procedure prescribes to consider the random variable $\bar{N} = N/N_{50\%}(\Delta\bar{\sigma})$, which is obtained by normalizing the specimen fatigue life $N$,

with respect to the mean fatigue life $N_{50\%}$ obtained from each sample subjected to a certain stress range $\Delta\bar{\sigma}$. In this way, the Weibull CDF of failure probability is made independent of the stress range. Therefore, the shape and scale parameters, $\bar{\alpha}$ and $\bar{\beta}$ become constant with respect to the stress range as well. Finally, the failure probability is expressed by the following equation:

$$P(\bar{N}) = 1 - e^{\left[-\left(\frac{\bar{N}}{\bar{\beta}}\right)^{\bar{\alpha}}\right]} \tag{3}$$

The best-fitting parameters $\bar{\alpha}$ and $\bar{\beta}$ can be assessed, according to [35], by the Maximum Likelihood Method (MLM), thanks to its high-reliability and flexibility.

Once that the fitting parameters are obtained, it is suitable to perform goodness-of-fit statistic tests, which detect the discrepancy between the estimated adopted distribution and the empirical one. In fact, these statistical tests measure the distance between observed and expected values, describing the level of adherence of the selected CDF to the sample, which is represented by the fatigue test data. Once that the selected cumulative distribution is checked, it is possible to compute S-N curves for a different probability of failure, $P$, by using the following expression according to [33]:

$$\log N = \bar{\beta}[-\ln(1 - P)]^{\bar{\alpha}} \left(\frac{\Delta\sigma_{0,50\%}}{\Delta\sigma}\right)^n \tag{4}$$

where $\Delta\sigma_{0,50\%}$ and $n$ are the best-fitting parameters of the mean life curve, whereas $\bar{\alpha}$ and $\bar{\beta}$ are the two parameters of Weibull distribution. Hence, Equation (4) makes it possible to obtain the number of cycles to failure for different values of stress range, given a certain probability of failure.

Figures 10–12 show the estimated CDFs for the three different specimens sizes, which are plotted against the experimental data. The cumulative relative frequency is evaluated according to Hazen's plotting position:

$$P_{\text{emp}} = \frac{i - 0.5}{m} \tag{5}$$

where $i$ is the position of the considered element, while $m$ is the sample size.

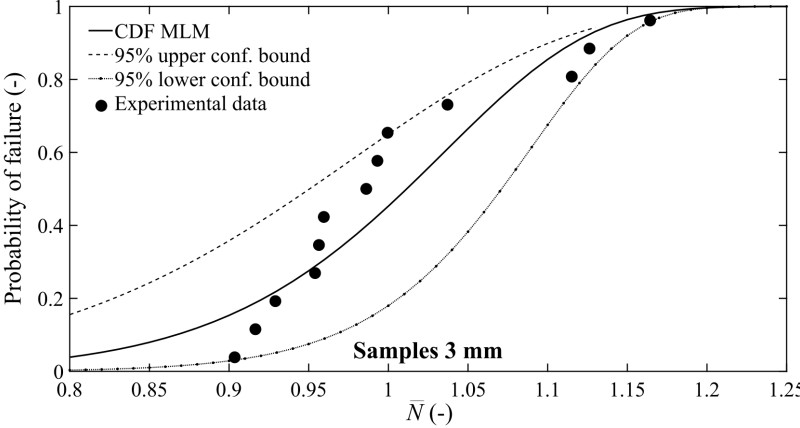

**Figure 10.** CDF for the specimens of 3 mm in diameter.

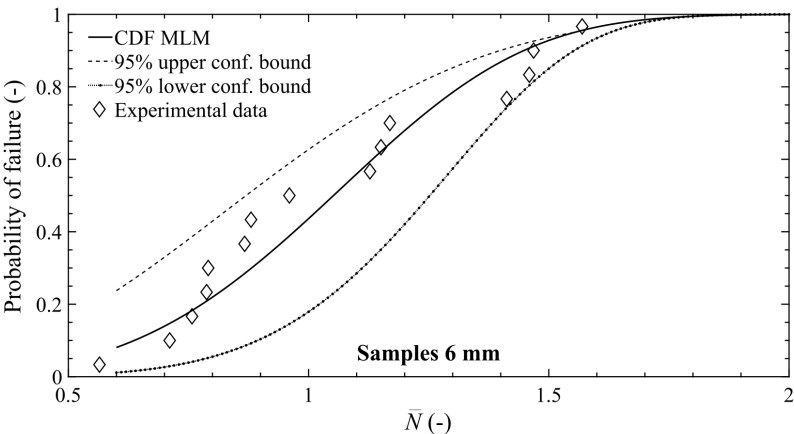

**Figure 11.** CDF for the specimens of 6 mm in diameter.

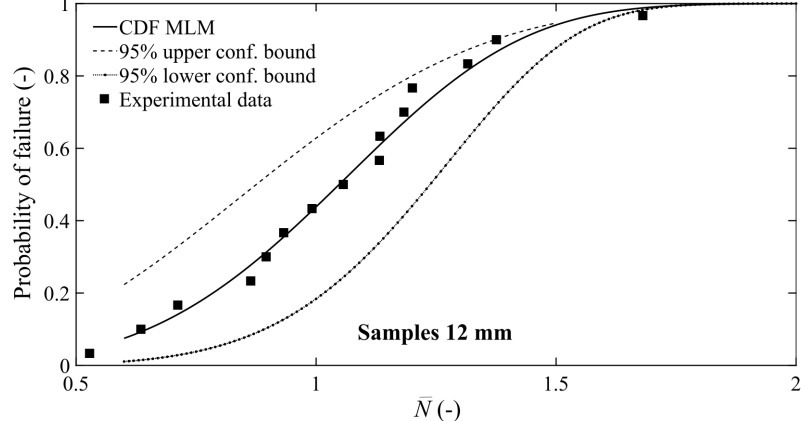

**Figure 12.** CDF for the specimens of 12 mm in diameter.

Table 4 reports the estimated two parameters of Weibull distribution for the three different specimen sizes.

**Table 4.** Assessment of shape and scale parameters from the experimental data.

| Diameter | $\bar{\alpha}$ | $\bar{\beta}$ |
|---|---|---|
| 3 mm | 12.2091 | 1.0423 |
| 6 mm | 3.7579 | 1.1596 |
| 12 mm | 3.9214 | 1.1508 |

The evaluation of Weibull CDF adherence to the experimental data is carried out by means of four different goodness-of-fit statistics tests: the $\chi^2$ test, the Anderson–Darling (A-D) test, the Cramer-von Mises (C-vM) test, and Kolmogorov–Smirnov (K-S) test. Table 5 reports the values obtained by each goodness-of-fit statistics test for the specimens of 3 mm in diameter. In addition, the critical values obtained for the 5% significance level are reported. Analogously, Tables 6 and 7 report the goodness-of-fit statistics for the specimens of 6 mm and 12 mm, respectively. In all cases, the estimated values for each goodness-of-fit (GoF) statistics tests and for each specimen size are smaller than the critical ones. Therefore, although further statistical investigation could provide a more suitable distribution to adopt, the null hypothesis for the two-parameter Weibull distribution is not rejected at the 5% significance level.

**Table 5.** Goodness-of-fit statistics tests results for the specimens of 3 mm in diameter.

| GoF Statistics Tests | $\chi^2$ | A-D | C-vM | K-S |
|---|---|---|---|---|
| Actual values | 7.792 | 0.701 | 0.143 | 0.242 |
| Critical values | 7.815 | 0.728 | 0.214 | 0.316 |

**Table 6.** Goodness-of-fit statistics tests results for the specimens of 6 mm in diameter.

| GoF Statistics Tests | $\chi^2$ | A-D | C-vM | K-S |
|---|---|---|---|---|
| Actual values | 7.000 | 0.475 | 0.077 | 0.168 |
| Critical values | 7.815 | 0.731 | 0.215 | 0.331 |

**Table 7.** Goodness-of-fit statistics tests results for the specimens of 12 mm in diameter.

| GoF Statistics Tests | $\chi^2$ | A-D | C-vM | K-S |
|---|---|---|---|---|
| Actual values | 0.600 | 0.156 | 0.019 | 0.107 |
| Critical values | 7.815 | 0.731 | 0.215 | 0.338 |

Eventually, Figures 13–15 show the obtained probabilistic S-N curves for the three different specimen sizes. Note that the S-N curves are assessed with the two-parameter Weibull distribution in correspondence of the 10%, 50%, and 90% probability of failure respectively.

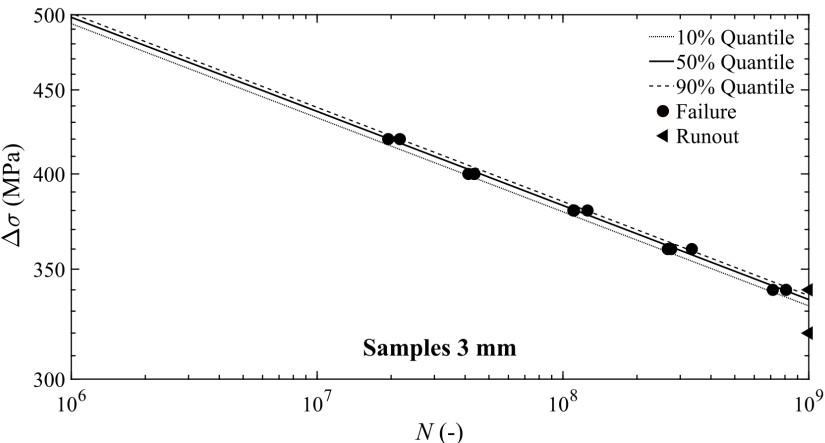

**Figure 13.** P-S-N curves for the specimens of 3 mm in diameter.

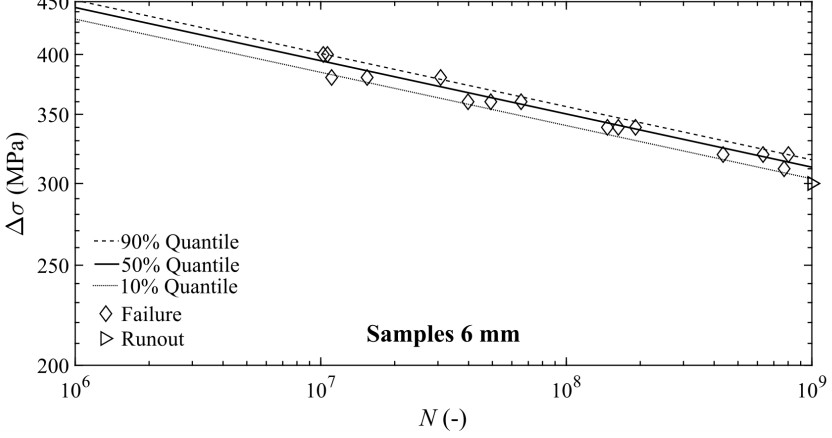

**Figure 14.** P-S-N curves for the specimens of 6 mm in diameter.

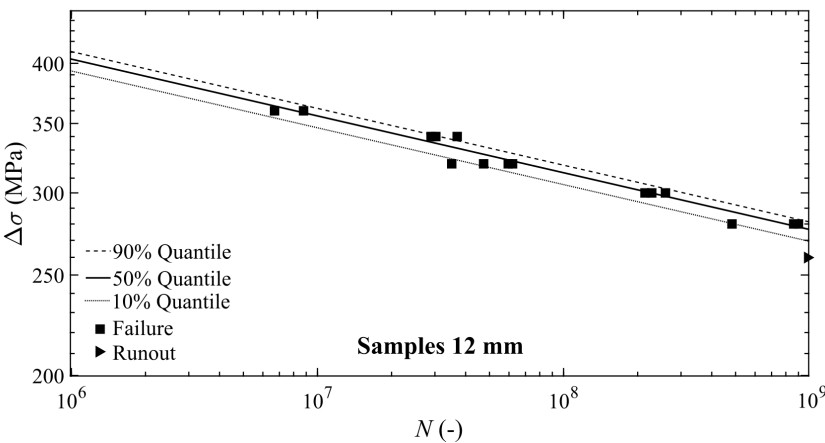

**Figure 15.** P-S-N curves for the specimens of 12 mm in diameter.

### 3.3. Theoretical Interpretation of the Specimen-Size Effect on Fatigue Resistance

Almost two decades ago, the concept of fractality was successfully exploited by Carpinteri to explain the size effect on Fracture Mechanics parameters [36–39]. In this framework, the material disorder due to the presence of a random distribution of flaws, inclusions, and micro-cracks [40], is accounted for adopting a fractal, rather than Euclidean, geometrical model [41,42]. In the following, the concept of fractality is adopted to provide a theoretical explanation for the decrease in fatigue resistance with the specimen size. From a practical point of view, the disorder due to inherent defects is taken into account by assuming that the reacting section, or ligament, of a disordered material is a lacunar fractal set with non-integer dimension lower than 2. Consequently, the following scaling law for the stress range can be assumed [23,43–45]:

$$\Delta\sigma = \Delta\sigma^* b^{-d_\sigma} \tag{6}$$

where $\Delta\sigma^*$ is the fractal stress range, with physical dimensions given by $[F][L]^{-(2-d_\sigma)}$, whereas $d_\sigma$ is the dimensional decrement of the ligament due to the presence of cracks and voids distribution, which ranges from 0 up to 0.5. Anagously, a similar scaling law for the intercept with the ordinate axis of the mean S-N curve, $\Delta\sigma_{0,50\%}$, can be put forward:

$$\Delta\sigma_{0,50\%} = \Delta\sigma^*_{0,50\%} b^{-d_\sigma} \tag{7}$$

where the term $\Delta\sigma^*_{0,50\%}$ is the coefficient of the power-law for the median fractal Wöhler's curve. Substituting Equation (7) in Equation (4), yields to a vertical downward translation of P-S-N curves for increasing specimen sizes, i.e., the larger the specimen dimension, the lower the fatigue strength. Thus, a set of P-S-N curves as a function of the specimen size $b$ can be obtained with the following analytical expression:

$$\log N = \bar{\beta}(b)[-\ln(1-P)]^{\bar{\alpha}(b)} \left(\frac{\Delta\sigma^*_{0,50\%}}{\Delta\sigma}\right)^n b^{-n d_\sigma} \tag{8}$$

The terms $\bar{\beta}(b)$ and $\bar{\alpha}(b)$ are the scale and shape parameters, which are not constant with the specimen size, as shown in Table 4. In other words, the scatter of experimental data is not constant with the structural size, but it decreases with it for a given stress range. Figure 16 plots the experimental data obtained by specimens of 3 mm, 6 mm, and 12 mm in diameter, together with the fitting median P-S-N curves expressed by Equation (8) in correspondence with a 50% failure probability. From a comparison between Figure 16 and Figures 13–15 it emerges clearly that the decrease in fatigue resistance with the specimen size is more pronounced than the scatter of experimental data, for the whole investigated dimensional range.

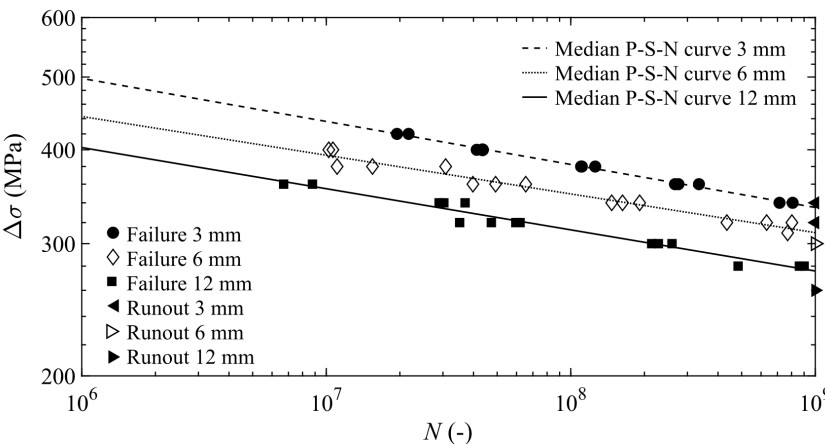

**Figure 16.** Specimen-size dependent median P-S-N curves in VHCF regime.

On the other hand, if the fractal stress range is used to represent experimental results, i.e. substituting Equation (6) in Equation (8),the set of specimen-size dependent P-S-N curves collapse onto a single fractal specimen-size independent P-S*-N curve, which is represented by the following equation:

$$\log N = \bar{\beta}^* \left[ -\ln(1-P) \right]^{\bar{\alpha}^*} \left( \frac{\Delta\sigma^*_{0,50\%}}{\Delta\sigma^*} \right)^n \tag{9}$$

The terms $\bar{\beta}^*$ and $\bar{\alpha}^*$ entering Equation (9) are the scale and shape parameters of the fractal P-S*-N curves, which are specimen-size independent.

The five free-parameters to be estimated in Equations (8) and (9) are: the dimensional decrement, $d_\sigma$; the coefficient of the power-law for the median fractal Wöhler's curve, $\Delta\sigma^*_{0,50\%}$; the exponent $n$; and the Weibull distribution coefficients $\bar{\beta}^*$ and $\bar{\alpha}^*$. The estimation is carried out in two steps. First, the dimensional decrement $d_\sigma$, the coefficient $\Delta\sigma^*_{0,50\%}$, and the exponent $n$ are assessed from a non-linear regression analysis of the experimental data. In this way the median fractal Wöhler's curve is obtained, in correspondence of $d_\sigma$ equal to 0.15, and the fractal stress-range shows the anomalous physical dimensions $[F][L]^{-1.85}$. Subsequently, if the experimental data are renormalized with respect to the fractal stress range, analogously to what was done in Section 3.2, the invariant parameters $\bar{\beta}^*$ and $\bar{\alpha}^*$ of the Weibull distribution are obtained, together with the fractal P-S*-N curves shown in Figure 17.

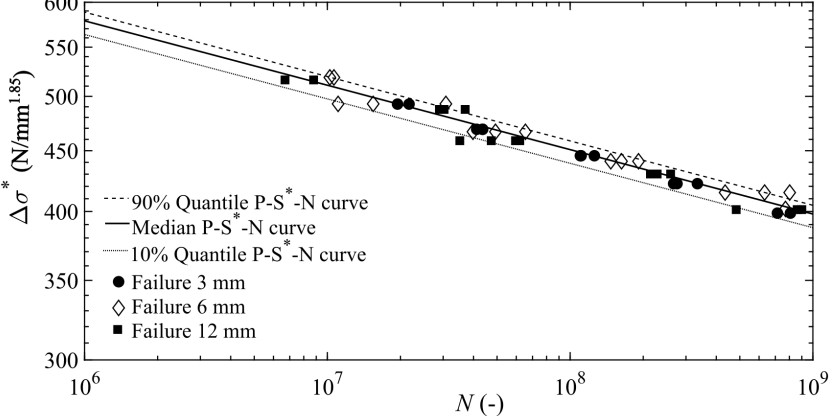

**Figure 17.** Fractal median P-S*-N curve in VHCF regime.

Eventually, in Table 8 the estimated five free-parameters are reported, as well as the coefficient of determination of the best-fitting analysis.

**Table 8.** Best-fitting of free-parameters entering Equations (8) and (9).

| $\Delta\sigma^*_{0,50\%}$ (N mm$^{-1.85}$) | $d_\sigma$ | $n$ | $R^2$ | $\bar{\alpha}^*$ | $\bar{\beta}^*$ |
|:---:|:---:|:---:|:---:|:---:|:---:|
| 1221 | 0.15 | 18.5 | 0.966 | 3.8522 | 1.1428 |

## 4. Conclusions

In the present investigation, fully reversed ultrasonic fatigue tests at a loading frequency of 20 kHz were performed on hourglass specimens made of EN-AW6082-T6 aluminum alloy up to $10^9$ cycles. Furthermore, three different dimensions of specimens were considered in order to investigate the specimen-size effects in the VHCF regime. The geometry of the specimens was designed by using a commercial finite element code. Thus, strain gauge calibration was carried out before the tests to check the stress distribution of the designed hourglass specimens. From the obtained experimental results, it emerges clearly that a decrease in the fatigue resistance with the specimen size is expected. Furthermore, the fractography analysis shows that the fatigue crack nucleation always originated from the subsurface. Subsequently, P-S-N curves were obtained for the three different specimen sizes, which show that the specimen size effect prevails on the scatter of experimental data. Scale-invariant P-S$^*$-N probabilistic fractal Wöhler curves are obtained by exploiting the concept of lacunar fractality, and the decrease in the fatigue life with the specimen size is theoretically justified and assessed. Although the extrapolation of the ultrasonic test result to structural components is not yet unanimously agreed, and the influence of loading frequency is still controversial, the P-S$^*$-N probabilistic fractal Wöhler curves can be used to provide objective values of the fatigue strength of full-size components subjected to ultrasonic VHCF regimes.

**Author Contributions:** conceptualization, S.I., F.M. and A.C.; methodology, S.I., F.M. and A.C.; investigation, F.M. and S.I.; data curation, F.M.; writing–original draft preparation, S.I. and F.M.; writing–review and editing, S.I. and A.C. All authors have read and agreed to the published version of the manuscript.

**Funding:** This research received no external funding.

**Institutional Review Board Statement:** Not applicable.

**Informed Consent Statement:** Not applicable.

**Data Availability Statement:** Data are contained within the article.

**Acknowledgments:** The authors wish to thank the company Italsigma$^{®}$ for the collaboration and its kind availability. SEM fractography was carried out at the J-Tech center of the Politecnico di Torino, whose collaboration is deeply acknowledged.

**Conflicts of Interest:** The authors declare no conflict of interest.

## Abbreviations

The following abbreviations are used in this manuscript:

| | |
|---|---|
| VHCF | Very High Cycle Fatigue |
| SLM | Selective Laser Melting |
| AISI | American Iron and Steel Institute |
| P-S-N | Probabilistic Stress-Life |
| UFTM | Ultrasonic Fatigue Testing Machine |
| SEM | Scanning Electron Microscope |
| CDF | Cumulative Distribution Function |
| MLM | Maximum Likelihood Method |

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
