# Peer review of "Experimental Evidence of Specimen-Size Effects on EN-AW6082 Aluminum Alloy in VHCF Regime"

_applsci, doi:10.3390/app11094272_

Round 1

Reviewer 1 Report

Really nice work! I made my comments in the PDF-document.

Author Response

We thank the reviewer for positive feedback and for the detailed comments, which have been answered in the Word attachment, and that allowed to improve the manuscript.   Please see the attachment

Reviewer 2 Report

The article is valuable for the science . I propose publication in present form.

Please correct the year of publication: 36. Carpinteri, A .; Chiaia, B .; Cornetti, P. A Scale-Invariant Cohesive Crack Model for Quasi-Brittle Materials. 401 Eng. Fract. Down. 2002, 69 (2), 207–217. DOI: 10.1016 / S0013-7944 (01) 00085-6

Author Response

We thank the reviewer for positive feedback and for the careful reading of the paper. The reference has been corrected in the manuscript as suggested.

Reviewer 3 Report

This paper is original and very interesting.  It is clearly and well written. Research design is appropriate; the methodology is adequately described, while the results and conclusions are clearly presented.

I think this paper is original work and may be published in the present form.

Author Response

We thank the reviewer for his positive feedback.